# Friend or Foe? Mining Suspicious Behavior via Graph Capsule Infomax Detector against Fraudsters

## ABSTRACT

Anomaly detection on graphs has recently attracted considerable attention due to its broad range of high-impact applications, including cybersecurity, financial transactions, and recommendation systems. Although many efforts have thus far been made, how to effectively handle the high inconsistency between users' behavior and labels, a fundamental issue in anomaly detection, has not yet received sufficient concern. Moreover, the inconsistency problem is hard to investigate and even deteriorates the performance of anomaly detectors. To this end, we propose a novel graph self-supervised learning framework, Capsule Graph Infomax (termed CapsGI), to overcome the inconsistency of anomaly detection. Inspired by the recent advances of capsules on images, we explore another possibility of reforming the node embedding by capsule ideas to represent the unique node's properties. Concretely, by disentangling heterogeneous factors underlying each node representation, we can establish node capsules such that their representation can reflect intrinsic node properties. To strengthen the connection among normal nodes, CapsGI further represents the part-whole contrastive learning between lower-level capsules (part) and higher-level capsules (whole) by explicitly considering the context graph relations. Extensive experiments on multiple real-world datasets demonstrate that our model significantly outperforms state-of-the-art models.

## CCS CONCEPTS

• **Computer systems organization** → **Embedded systems**; *Redundancy*; Robotics; • **Networks** → Network reliability.

## KEYWORDS

Anomaly detection, Graph neural networks, Self-supervised learning, Capsule networks

**ACM Reference Format:**
Anonymous Author(s). 2018. Friend or Foe? Mining Suspicious Behavior via Graph Capsule Infomax Detector against Fraudsters. In *Proceedings of Make sure to enter the correct conference title from your rights confirmation emai (Conference acronym 'XX)*. ACM, New York, NY, USA, 10 pages. https://doi.org/XXXXXXX.XXXXXXX

## 1 INTRODUCTION

As Internet economics and beyond networks thrive, they also incubate various fraudulent activities, e.g., calling card and telecommunications fraud [4], fake accounts on social networks [3], network intrusion anomaly detection [5], and commit download fraud [6]. Roughly speaking, fraudsters disguise themselves as normal users to hide in the crowd (so-called spamouflage), so as to bypass the anti-fraud system and disperse disinformation [7]. To circumvent those suspicious behaviors, graph-structured approaches for anomaly detection have become a promising development in both industrial communities and academic [24, 31, 42], as graphs effectively describe the correlations among users or objects that participate in the fraudulent activities [42].

Along with the recent advances in graph neural networks (GNNs) [23, 43, 52], whose breakthrough performance is highly related to the representation of nodes, numerous endeavors seek to adopt GNNs for anomaly detection [50, 53, 54]. Previous works aim to promote identifying quality by studying promising GNN encoders to spot abnormal instances from the massive data. In this paper, we concentrate on detecting abnormal users in the real-world graph structured data, which is relevant to the Web and social networks.

Albeit the remarkable performance, existing GNN-based models detecting anomalies are not trivial due to the diversity of anomalies. Conceivably, savvy fraudsters can connect to normal users and obtain information from neighbors by GNNs as camouflage, which helps the fraudsters prevent themselves from being spotted by anomaly detectors [28]. Such camouflage will cause a severe inconsistency between users' behavior and label semantics in anomaly detection. However, the dominant algorithms focus less on the inconsistency problem when designing specific GNN-based detectors, which may deteriorate their performance. To better clarify this issue, **Figure 1** illustrates an example of an opinion fraud graph, where the green entities denote normal users and the red ones represent fraudsters. We can observe an interesting phenomenon as follows:

$(u_1, u_3)$ *has same label with behavior inconsistency, while* $(u_1, u_2)$ *has similar behavior patterns, but label inconsistency.* In Figure 1(b), suppose that $u_1$ focuses on hotel reviews, while $u_3$ is absorbed in film reviews. Their review contents (features) are far from each other as they are associated with different themes. Nevertheless, they both are normal users with normal conduct comments from a global perspective. Thus $(u_1, u_3)$ behave differently from each other, even if they share the same label. Concurrently, in Figure 1(c), those crafty fraudsters (e.g., $u_2$) seek to connect with neighbor users (e.g., $u_1$) and learn information from GNNs (e.g., hotel reviews) to hide in the normal users. That makes anomaly detectors confused and hard to tell $u_2$ whether a *friend or foe*, resulting in a situation where the distance in $(u_1, u_3)$ appears farther away than $(u_1, u_2)$. **Such inconsistency between users' behaviors and labels** will put the GNN-based anomaly detectors in a dilemma: to

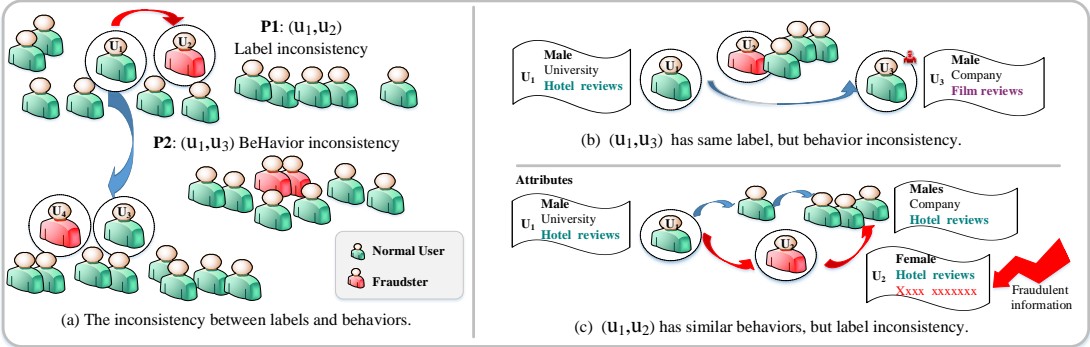

**Figure 1: Motivation of our work. (a) An illustration of the inconsistency problem between the behavior patterns and the label semantics in anomaly detection. Take the opinion fraud (e.g., spam reviews) detection problem as an example.(b) $u_1$ and $u_3$ are normal users but perform different behaviors (e.g., $u_1$ focuses on hotel reviews, while $u_3$ is absorbed in film reviews), resulting in that $(u_1, u_3)$ has the same label but behavior inconsistency. (c) Fraudster $u_2$ adjusts his behaviors, carrying the intention of fraudulent information when propagation, to gloss over suspicious behaviors and hide among normal users (e.g., fraudster $u_2$ would pretend to like hotel reviews to imitate the normal user $u_1$). Such behavior leads to the phenomenon that $(u_1, u_2)$ has similar behaviors but label inconsistency.**

learn the intrinsic node properties or to strengthen the connection between normal nodes. Although previous works [26, 42] also have noticed similar challenges, their solutions either fail to fit the fraud detection problems or break the end-to-end learning fashion of GNNs, requiring more training times and resources. To this end, we propose a simple yet effective graph anomaly detection approach based on capsule networks, called Capsule Graph Infomax (termed CapsGI), that focuses on two specific perspectives (local and global) to alleviate the impact of the inconsistency problem.

**The first is the local perspective (P1)**: We disentangle the node representations to better tell the difference between normal users and fraudsters for anomaly detection. Inspired by the recent progress of capsules in images [12, 33, 47], capsule neural networks (CapsNets) have proved their effectiveness in capturing atom features on image data [13], and thus can retain more rich information. In order to mine the users' unique features against fraudsters, we build node capsule representation by disentangling heterogeneous factors behind the node embeddings, such that each node can be represented the intrinsic properties of normal users for better distinguishing from fraudsters.

**The second is the global perspective (P2)**: We should also strengthen the connection between normal nodes from a global perspective. The main reason is that considering the agility of real-world fraudsters [7], exactly capturing those camouflaged fraudsters and sweeping them all away is impractical. Moreover, another challenge in anomaly detection lies in the insufficient labeled data. Anomaly data will finally be mixed with the normal ones in real-world applications as time goes by. Motivated by the recent deep graph infomax (DGI) [37] that relies on maximizing mutual information between each node representation and corresponding global graphs, we propose a new graph SSL scheme, Capsule Graph Infomax (GapsGI), which inherits the strength of DGI for anomaly detection. Specifically, GapsGI further characterizes the

part-whole contrastive learning between lower-level capsules (local) and higher-level capsules (global) by explicitly considering the context graph relations. Additionally, GapsGI is particularly attractive in inferring the context of the local part connected to the global distribution of nodes. This strategy enables GapsGI not only to capture the distribution of nodes but also to retain more intrinsic and unique node information. Summarily, we highlight our key contributions as follows:

- We propose a novel framework called Capsule Graph Infomax (CapsGI), which employs capsule ideas to address the inconsistency issue for anomaly detection. To the best of our knowledge, we are the first to combine capsules with self-supervised learning (SSL) to conduct anomaly detection on graphs.

- We demonstrate the effectiveness of considering the entanglement of latent factors against camouflaged fraudsters, and the stronger connection in modeling part-whole contrastive relationships on the normal nodes.

- Extensive experiments are conducted on multiple public real-world datasets, showing that our approach achieves competitive performance compared with existing graph approaches.

## 2 RELATED WORK

### 2.1 Graph-based Anomaly Detection

**Graph-based Anomaly Detection.** Anomaly data or behaviors are ubiquitous in a wide range of realistic scenes [25]. Generally speaking, anomalies not only deteriorate the data quality but also increase the model complexity. Early researches detect anomalies via iterative learning [20, 39, 40], dense block identification [14, 34, 35], or belief propagation on graphs [2, 32]. Afterward, graph-based anomaly detection [24, 31, 42] has drawn great attention, especially the recently emerged GNN-based approaches [50, 53, 54], which aim to learn expressive node representations with the goal of identifying abnormal instances in the embedding space [42]. For instance,

CARE-GNN [7] leverages reinforcement learning to find the optimal amounts of neighbors, so as to enhance the GNN aggregation process. GeniePath [27] studied the problems of GNNs in identifying meaningful receptive paths to guide the receptive paths. SemiGNN [38] used a hierarchical attention mechanism to correlate different neighbors better. ADMoE [51] leverages Mixtureof-experts (MoE) architecture to encourage specialized and scalable learning from multiple noisy sources. Unlike previous works, we attempt to explore another possibility of disentangling the node representation to mine heterogeneous factors by capsule ideas.

## 2.2 Graph Self-supervised Learning

Self-supervised learning (SSL) is a new learning paradigm that aims to learn a neural network of the unsupervised data itself without using human-annotated labels [53]. Existing graph contrastive learning (GCL) is a class of SSL that learns to distinguish data by pulling similar nodes together and pushing dissimilar nodes away. For instance, GAE [18] reconstruct the one-hop or the multi-hop adjacency information between nodes. GRADATE [8] regards the original input graph as the frst view and adopts edge modifcation as graph augmentation technology to generate the second view. ACT [41] connects an optimaltransport-based discrepancy measure and graph-structurebased contrastive loss to leverage prior AD knowledge from a source graph as a joint learning scheme. DCI [42] suggests that decoupled training equipped with a proper SSL objective can be an alternative way for effective anomaly detection. DGI [37] obtains node embeddings by leveraging local mutual information maximization across the graph's patch representations. Objectives of these methods are effective in GCL, while our work differs from the above methods. We contrast the local and the global representations in the form of capsules to overcome the inconsistency issue for anomaly detection.

## 2.3 Capsule Neural Networks

Recently, capsule neural networks (CapsNets) [12, 13, 33, 47, 52] have been actively used in many fields, such as action recognition [9], generative models [16], and health monitoring [1]. They use groups of neurons (named capsules) to represent entity features. To handle graphs under heterophily, NCGNN [48] adopts the concept of capsules, so that each node capsule adaptively aggregates advantageous capsules and restrains irrelevant messages. CapsGNN [45] utilizes capsule ideas to address the weakness in existing GNN-based graph embeddings algorithms. As for our work, we use the capsule ideas combined with self-supervised learning to model part-whole graph relationships, so as to strengthen the connection between normal nodes.

## 3 PRELIMINARY

### 3.1 Notations and Definition

*Definition 1: (Graph).* Let $\mathcal{G} = (\mathcal{V}, \mathcal{E}, \mathbf{A}, \mathbf{X})$ be a graph, where $\mathcal{V} = \{v_i\}_{i=1}^{N}$ ($|\mathcal{V}| = N$) is a set of nodes, and $\mathcal{E} = \left[ e_{i,j} \right] \in \mathbb{R}^{N \times N}$ denotes the corresponding edges. The neighborhood of a node $v_i$ is denoted as $\mathcal{N}(v_i) = \{v_j \in \mathcal{V} \mid e_{i,j} \in \mathcal{E}\}$. $\mathbf{X}$ represents the feature matrices of nodes. The topology of the graph is represented as an adjacency matrix $\mathbf{A} \in \mathbb{R}^{N \times N}$, where

$$\mathbf{A}_{i,j} = \begin{cases} 1, e_{i,j} \in \mathcal{E} \\ 0, e_{i,j} \notin \mathcal{E} \end{cases} \quad . \tag{1}$$

*Definition 2: (**Anomaly Detection Graph**).* An anomaly detection graph refers to a graph where nodes or edges are associated with their own features (e.g., attributes). The feature matrices of nodes and edges are represented as $\mathbf{X}_{\mathcal{N}} \in \mathbb{R}^{N \times d_{node}}$ and $\mathbf{X}_{\mathcal{E}} \in \mathbb{R}^{m \times d_{edge}}$ respectively. Although the nodes usually include users and objects in the anomaly detection graph, we only care about detecting anomaly users. On a parallel note, in a more common scenario where only nodes have features, we use $\mathbf{X} \in \mathbb{R}^{N \times d}$ to denote the node feature matrix for simplicity, and denote the anomaly detection graph as $\mathcal{G} = (\mathcal{V}, \mathcal{E}, \mathbf{A}, \mathbf{X}, \mathcal{Y})$ (**Figure 2a**), where $\mathcal{Y}$ is a set of labels on nodes.

*Definition 3: (**Inconsistency**).* Similar behavior of users, but have opposite labels. Or users with the same label perform far different behaviors.

### 3.2 Problem Definition

**Problem 1.** *Anomaly Detection on a Graph.*
- **Given:** Given a ***partially*** labeled anomaly detection graph $\mathcal{G} = (\mathcal{V}, \mathbf{A}, \mathbf{X}, \mathcal{Y}^L)$, where $\mathcal{Y}^L$ is a set of partial labels on nodes. The partial nodes are associated with the label $y_i \in \{0, 1\}$, where $0$ denotes that the node is a normal user and $1$ represents a fraudster.
- **Goal:** The objective of anomaly detection is to learn a predictive function:

$$\mathcal{F} : \mathbb{R}^{N \times d} \to \mathbb{R}^{N \times 1} \tag{2}$$

We measure the degree of abnormality of a node by calculating its anomaly score, then we rank node anomaly scores in descending order. Finally, anomalies can be easily detected based on this ranking list.

### 3.3 GNN-based Anomaly Detectors

We detect the fraud entities in a graph by using the node representations. Hence, we need to first introduce the node representation learning by GNN-based anomaly detectors and how to work, including graph encoder and anomaly classifier, to perform our study.
**Graph Encoder.** Generally, the advanced graph-based anomaly methods almost all extend the existing GCN [19], GIN [46], GraphSAGE [11], or GAT [36] to tackle the problem of anomaly detection. They make efforts to design promising GNN encoders for better node representations. Concretely, GNNs utilize a neighborhood aggregation scheme [15] to learn a representation $h_i$ for each node $v_i$, and then perform $k$ rounds of neighbor aggregation. Formally, consider the general GNN framework in **Table 1**, we take GIN as an example to instantiate the GNN encoder in the anomaly problem. We initialize $h_i^{(0)} = x_{v_i}$. After $k$ rounds of aggregation, each node $v_i \in \mathcal{V}$ obtains its representation $h_i^{(k)} \in \mathbb{R}^d$, aggregated from their neighbors $\mathcal{N}_{v_i}$. The other three GNNs have similar neighborhood aggregation and can generalize to more datasets for anomaly detection.
**Anomaly Classifier.** To project the target node to the same embedding space, we instantiate an anomaly classifier based on a

**Table 1: Neighborhood aggregation schemes**

| Methods | Aggregation and combination functions for round $k(1 \leq k \leq K)$ |
|---|---|
| General GNN framework | $h_i^{(k)} = \text{COMBINE}^{(k)} \left( \left\{ h_i^{(k-1)}, \text{AGGREGATE}^{(k)} \left( \left\{ h_j^{(k-1)} : v_j \in N_{v_i} \right\} \right) \right\} \right)$ |
| GCN | $h_i^{(k)} = \Theta \left( \sum_{v_j \in N_{v_i} \cup \{v_i\}} \frac{1}{\sqrt{(|N_{v_i}|+1) \cdot (|N_{v_j}|+1)}} \cdot \mathbf{W}^{(k-1)} \cdot h_{v_j}^{(k-1)} \right)$ |
| GIN | $h_i^{(k)} = \Theta \left( (1 + \epsilon) \cdot h_i^{(k-1)} + \sum_{v_j \in N_{v_i}} h_j^{(k-1)} \right)$ |
| GraphSAGE | $h_i^{(k)} = \Theta \left( \mathbf{W}^{(k-1)} \cdot \left[ h_i^{(k-1)} \parallel \sum_{v_j \in N_{v_i}} h_j^{(k-1)} \right] \right)$ |
| GAT | $h_i^{(k)} = \Theta \left( \sum_{v' \in N_v \cup \{v\}} a_{i,j}^{(k-1)} \cdot \mathbf{W}^{(k-1)} \cdot h_j^{(k-1)} \right)$ |

linear mapping followed by a *partial supervised loss* function $\mathcal{L}_{PSL}$ to minimize the cross-entropy loss, i.e.,

$$\mathcal{L}_{PSL} = -\frac{1}{|\mathcal{Y}^{\mathcal{L}}|} \sum_{i=1}^{|\mathcal{Y}^{\mathcal{L}}|} \left[ (y_i \cdot \log p_i + (1 - y_i) \cdot \log(1 - p_i) \right] \quad (3)$$

$$p_i = \sigma \left( \mathbf{W}^T h_{v_i}^{(k)} + b \right) \quad (4)$$

where $p_i$ is the predicted suspicious score (i.e., abnormal score) of node $v_i$. $\sigma$ is the sigmoid activation function, $\mathbf{W}$ and $b$ are the parameters to be learned.

## 4 METHODOLOGY

For anomaly detection, we propose CapsGI, as shown in **Figure 2**, to remedy the inconsistency between behavior patterns and label semantics in anomaly detection. In what follows, we elaborate on two essential components (P1 and P2) of the proposed framework.

### 4.1 Disentangled Node Capsules

In most cases, highly complex interactions (e.g., common hobbies like reviews, and films) are involved in connecting each user in society. We hope to exploit these attributes to distinguish normal users from fraudsters in anomaly detection. However, the reality may be far from satisfactory. Aggregating the information of normal users by GNNs can help fraudsters integrate into the normal users as soon as possible, and thus disguise themselves from being spotted by anomaly detectors. Therefore, it is necessary to disentangle heterogeneous factors underlying each node to explore their intrinsic properties. Furthermore, fraudsters may have unique intentions (features) and are hard to be found by the existing graph methods. (e.g., they add some special characters to a fake review, which may help to bypass feature-based detectors and propagate their fraudulent information).

To address the above challenges, motivated by capsule ideas [12, 33], we propose disentangled node capsules (DNC) to describe the node embeddings. Specifically, we disentangle the latent factors of each node embedding and use the disentangled node representation to represent node capsules (**Figure 2b**). In this way, each node capsule is composed of multiple heterogeneous factors, and each factor describes a specific and inherent instantiation feature of the node (users), including the fraudsters' unique attributes. Formally, given $\mathcal{G} = (\mathcal{V}, \mathcal{E}, \mathbf{A}, \mathbf{X}, \mathcal{Y}^L)$, we define $\mathbf{X} = \{\mathbf{x}_1, \mathbf{x}_2, \cdots, \mathbf{x}_N\} \in \mathbb{R}^{N \times d}$ as node features. Assuming that there are $K$ capsule subspaces and each subspace has $\frac{h}{K}$ dimensions ($h$ is the final dimension after

disentanglement operations), we project each node from the scalar-based features into $K$ different capsule subspaces to obtain node capsule, represented by a pose matrix $\mathbf{z}_i \in \mathbb{R}^{K \times \frac{h}{K}}$ [13].

$$\mathbf{z}_i^{(l)} = \mathcal{M} \left( \mathbf{W}_k^{(l)} \mathbf{x}_i + \mathbf{b}_k \right), k = 1, 2, \cdots K \quad (5)$$

where $\mathbf{W}_k^{(l)} \in \mathbb{R}^{d \times \frac{h}{K}}$ and $\mathbf{b}_k \in \mathbb{R}^{\frac{h}{K}}$ are the learnable weights and bias with respect to the $k$-th capsule subspace. $\mathcal{M}(\cdot)$ is a nonlinear activation function. Although more sophisticated implementations of node disentanglement are possible [29], we use linear projection in our study attributed to its efficiency and remarkable performance. For simplicity, we reshape $\mathbf{x}_i^{(l)} \in \mathbb{R}^d$ to the vector format $\mathbf{z}_i^{(l)} \in \mathbb{R}^h$, where $d < h$. Then, we compute the length of the output vector of each node capsule to indicate the presence of a node via using the squash activation function [12]:

$$\mathbf{u}_i^{(l)} = squash(\mathbf{z}_i) = \frac{\|\mathbf{z}_i\|^2}{1 + \|\mathbf{z}_i\|^2} \frac{\mathbf{z}_i}{\|\mathbf{z}_i\|} \quad (6)$$

where $\mathbf{u}_i^{(l)} \in \mathbb{R}^h$ represents the lowest level of entities, such as an eye in a face. $\frac{\|\mathbf{z}_i\|^2}{1 + \|\mathbf{z}_i\|^2}$ is the scale by which the vector is scaled, and $\frac{\mathbf{z}_i}{\|\mathbf{z}_i\|}$ is to preserve unit vectorization. Note that the value of $\mathbf{u}_i^{(l)}$ indicates how much influence to the higher capsules in the next layer. Through this mapping scheme, we project the node features to the capsule subspace to take into account the heterogeneous factors behind each node. Meanwhile, disentangling ideas can help us preserve the inherent properties of each normal user against fraudsters over the whole graph.

### 4.2 Capsule Graph Infomax

Disentangled node capsules can get better node representations through neighborhood aggregation schemes by promising GNNs, for the following anomaly classifier. However, various limitations of the data distributions such as label imbalance [42] and scarce labels [10] may increase the distinguishing difficulty of anomaly detectors. Furthermore, normal users have the same label, but behavior inconsistency also leads to misjudgment of the detectors. In such situation, we need to seek a reliable solution that not only can narrow the distribution distance of normal users, but also strengthen the connection between users with the same label from a global perspective. Recently, self-supervised learning (SSL) [44, 49] has achieved considerable performance due to playing a significant role in dealing with scarce labels, especially DGI [37] that enables node representations to capture the global information of the entire graph and strengthen the connection among nodes. Additionally, since the normal users usually occupy the majority of the data, we can represent the whole graph to approximate the distribution of the normal users [42]. Based on the above analysis, we propose capsule graph infomax (CapsGI), a new graph SSL scheme that inherits the strength of DGI to enhance the connection among normal nodes. Concretely, CapsGI has two steps:

***Step(1): How to generate the higher-level graph capsules?***
We conduct a V-GNN that lower-level node capsules can vote for the higher-level graph capsules (**voting**). When multiple votes agree, a higher-level capsule that receives a cluster of similar votes becomes active (**routing**). This process can generate higher-level

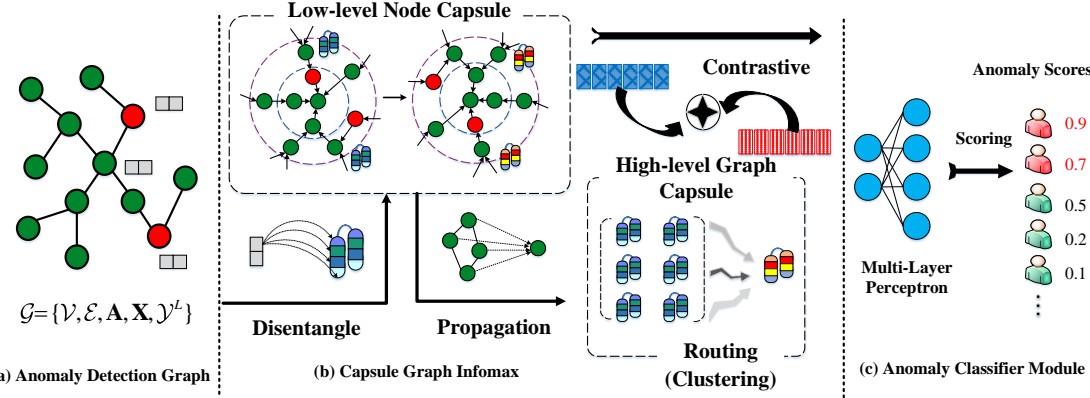

(a) Anomaly Detection Graph          (b) Capsule Graph Infomax          (c) Anomaly Classifier Module

**Figure 2: An overview of the proposed framework. (a) The input of anomaly detection graph $\mathcal{G} = (\mathcal{V}, \mathcal{E}, \mathbf{A}, \mathbf{X}, \mathcal{Y}^L)$. (b) The process of capsule graph infomax. We first disentangle heterogeneous factors behind the node embeddings to obtain the intrinsic node properties. Then, we propose CapsGI to narrow the distribution distance of normal users, but also strengthen the connection between users with the same label. (c) An anomaly Classifier to predict the suspicious score of nodes.**

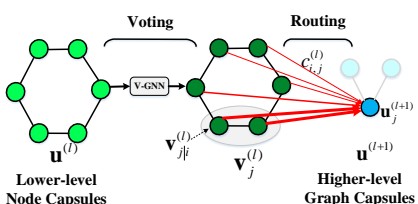

**Figure 3: Cluster by agreement.**

graph capsules that represent the whole graph. Formally, we denote the lower-level node capsules at layer $l$ as $\mathbf{u}^{(l)} \in \mathbb{R}^{N_l \times d_l}$ and the adjacency matrix as $\mathbf{A}^{(l)} \in \mathbb{R}^{N_l \times N_l}$. Our goal is to decide which capsules to activate in $\mathbf{u}^{(l+1)} \in \mathbb{R}^{N_{l+1} \times d_{l+1}}$ and how to assign each lower-level capsule $\mathbf{u}_i^{(l)}$ to make the connection for one higher-level capsule $\mathbf{u}_j^{(l+1)}$. In practice, we set $N_{l+1} < N_l$ in order to get coarser and coarser node representations. This process involves the voting and routing mechanism, as shown in **Figure 3.**.

**Voting:** Lower-level node capsules make a vote for the final prediction of higher-level graph capsules through V-GNN, which is formulated as:

$$\mathbf{v}_j^{(l)} = \text{V-GNN}(\mathbf{A}^{(l)}, \mathbf{u}^{(l)}) \tag{7}$$

where $\mathbf{v}_j^{(l)} \in \mathbb{R}^{N_l \times d_{l+1}}$. Specifically, $\mathbf{v}_{j|i}^{(l)} \in \mathbb{R}^{d_{l+1}}$ is the vote for $\mathbf{u}_j^{(l+1)}$ predicted by the node capsule $\mathbf{u}_i^{(l)}$. V-GNN is a general GNN framework that allows various choices of the network architecture without any constraint, which can be GIN, GCN, GIN, or GRAPHSAGE. Note that V-GNN is learned discriminatively to represent part-whole relationships by considering the information in $\mathbf{u}_i^{(l)}$.

**Routing:** A higher-level graph capsule, which denotes the whole distribution of all normal users in a graph, is activated when multiple predictions by lower-level capsules agree. In other words, **the dynamic routing of capsules is essentially a clustering process.** In this paper, each of these votes is then weighted by an

routing weight $c_{i,j}^{(l)}$ with which a part is assigned to a whole, where $c_{i,j}^{(l)} \geq 0$ and $\sum_{j=1}^{N_{l+1}} c_{i,j}^{(l)} = 0$ . Here, $c_{i,j}^{(l)}$ is iteratively updated using an "routing-by-agreement" mechanism such that each vote in $\mathbf{v}^{(l)}$ is routed to a capsule in $\mathbf{u}^{(l+1)}$ hat receives a cluster of similar votes (**Figure 3**). To iteratively search for the vote cluster, in each iteration we have,

$$\mathbf{u}_j^{(l+1)} = squash\left(\sum_i c_{i,j}^{(l)} \mathbf{v}_{j|i}^{(l)}\right) \tag{8}$$

$$c_{i,j}^{(l)} = \frac{exp(b_{i,j}^{(l)})}{\sum_k exp(b_{i,k}^{(l)})} \tag{9}$$

where squash($\cdot$) is to indicate the presence of an instance of class, where defined in Eq.(4). $b_{i,j}^{(l)}$ is initialized as $b_{i,j}^{(l)} = 0$. $\mathbf{u}_j^{(l+1)}$ ) is the predicted capsule $j$ in layer $l + 1$, representing a tight cluster of votes from layer $l$. Then we update $b_{i,j}^{(l)}$,

$$b_{i,j}^{(l)} = b_{i,j}^{(l)} + a_{i,j}^{(l)} \tag{10}$$

where $a_{i,j}^{(l)} = \mathbf{v}_{j|i}^{(l)} \cdot \mathbf{u}_j^{(l+1)}$ indicates the agreement between each vote and vote cluster. After the above operations, we get higher-level graph capsules $\mathbf{u}^{(l+1)}$ and the coarsened adjacency matrix $\mathbf{A}^{(l+1)}$ defined as:

$$\mathbf{A}^{(l+1)} = (c_{i,j}^{(l)})^T \mathbf{A}^{(l)} c_{i,j}^{(l)} \tag{11}$$

***Step(2): How to conduct the contrastive learning between lower-level node capsules and higher-level graph capsules?***

Finally, each higher-level graph capsule represents the vote of the whole lower-level node capsules. In this way, the same label of users (node capsules) tends to be clustered into the same cluster (graph capsule). After clustering, we compute the graph-level representation $\mathbf{s}$ for each graph capsule to summarize how the majority in the whole graph, i.e.,

$$\mathbf{s} = \sigma \left( \frac{1}{N_{l+1}} \sum_{j=1}^{N_{l+1}} \mathbf{u}_j^{(l+1)} \right) \quad (12)$$

The loss of self-supervised learning in graph capsule infomax can be defined as follows:

$$\mathcal{L}_{CapsGI}^k = \\ -\frac{1}{2n} \sum_{i=1}^{n} \left( \mathbb{E}_{\mathcal{G}} \log \mathcal{D} \left( \mathbf{u}_i^{(L)}, \mathbf{s} \right) + \mathbb{E}_{\tilde{\mathcal{G}}} \log \left( 1 - \mathcal{D} \left( \tilde{\mathbf{u}}_i^{(L)}, \mathbf{s} \right) \right) \right) \quad (13)$$

where $\mathcal{D}$ is a discriminator that outputs the affinity score of each local-global (i.e., node-graph) pair. Similar to DGI, graph $\tilde{\mathcal{G}}$, generated by a row-wise shuffling of the initial feature matrix $\mathbf{X}$, provides the node representation $\tilde{\mathbf{u}}_i^{(L)}$ that can be paired with the graph representation $\mathbf{s}$ as a negative sample. The final loss function of CapsGI is the average of the losses of the $K$ graph capsules, i.e.,

$$\mathcal{L}_{CapsGI} = \frac{1}{K} \sum_{k=1}^{K} \mathcal{L}_{CapsGI}^k \quad (14)$$

## 4.3 Model Optimization

Consequently, we unify the anomaly detection task and self-supervised task into a primary&auxiliary learning framework, where the former is the primary task and the latter is the auxiliary task. The joint learning objective is defined:

$$\mathcal{L}_{LOSS} = \mathcal{L}_{PSL} + \alpha \mathcal{L}_{CapsGI} \quad (15)$$

where $\alpha$ is a hyperparameter that controls the magnitude of the SSL task. It should be noted that we jointly optimize the two tasks throughout the training. And **why choose capsule representation?** The answer can be found in **Appendix A**.

## 5 EXPERIMENTAL EVALUATION

**Table 2: Statistics of the datasets.**

| Dataset | #Users(% normal, abnormal) | #Edges | #Objects |
|---------|---------------------------|--------|----------|
| Reddit | 10,000 (96.34%, 3.66%) | 78,516 | 984 |
| Alpha | 3,286 (61.21%, 38.79%) | 24,186 | 3,754 |
| Wikipedia | 8,227 (97.36%, 2.64%) | 18,257 | 1,000 |
| Amazon | 27,197 (91.73%, 8.27%) | 52,156 | 5,830 |

## 5.1 Experimental Settings

**Datesets.** We utilize four real-world benchmark datasets to study the inconsistency problem in the anomaly detection task. Detailed statistics about these datasets are listed in **Table 2**.

- **Reddit** [1] [22]: This public dataset is a user-subreddit graph with 672,447 interactions, which *contains ground-truth labels of banned users from Reddit.*
- **Alpha** [2] [21]: This public dataset is a user-user trust graph of Bitcoin. Only 214 users in this dataset are labeled.

---

[1]http://files.pushshift.io/reddit/
[2]http://www.bitcoin-otc.com and http://www.btcalpha.com

- **Wikipedia** [3] [22]: This public dataset is an editor-page graph with 157,474 interactions, which *contains public ground-truth labels of banned users.*
- **Amazon** [4] [30]: This public dataset is a user-product graph. Ground truth is defined by using votes containing malicious behavior. Only 278 users in this dataset are labeled.

Specifically, following the previous work [42], we extracted Amazon from a large user-product graph [30], and then use METIS [17] to divide the Amazon dataset into 20 sub-graphs, which preserves the original graph structure within the sampled dataset as much as possible. The graphs used in our experiments are unweighted, where the edge represents a user has ever interacted with an object.

**Baseline Methods.** We compare CapsGI with two categories of baselines, whose detailed introduction can be found in **Related Works**. **GNN-based methods**: We select GCN [19], GIN [46], Graph-SAGE [11], GAT [36], GeniePath [27], Semi-GNN [38], CARE-GNN [7], and ADMoE [51]. **Self-supervised learning for anomaly detectors**: We compare GRADATE [8], ACT [41], GAE [18], DGI [37], DCI [42] to show the superiority of our proposed CapsGI.

**Evaluation Metrics.** In practice, the real-world datasets of anomaly detection have imbalanced classes, that is, normal users dominate during the training procedure. However, we focus more on fraudsters, and all the predicted suspicious scores tend to be small, which makes it difficult to set a proper threshold for classifying the fraudsters and the normal users. So we adopt the widely used metric AUC [7] which is computed based on the relative ranking of prediction probabilities of all instances to eliminate the influence of imbalanced classes. For a fair comparison, we conduct a 10-fold evaluation on Reddit, Alpha and Wikipedia. Specially, we conducted a 5-fold evaluation on Amazon since this dataset has limited labeled fraudsters.

**Parameter Settings.** For all GNNs-based models, we set a learning rate (0.01), optimizer (Adam), and adopt input feature dimension (64), GNN layers (3), and node representation dimension (128). For different SSL schemes, we unify their backbones as the GIN's encoder. For GCN, GIN, GraphSage, Semi-GCN, CARE-GNN, AD-MoE, GAE, DGI, GRADATE, ACT, and DCI, we use the source code provided by their authors. For GAT and GeniePath, we use the open-source implementation [5]. We modified these codes to make them adapt to our tasks. For the classification, we record the best testing result after 100 epochs on each fold, then report the best AUC score over different folds.

## 5.2 Overall Comparison

To demonstrate the performance of CapsGI, we compare it with state-of-the-art anomaly detectors. The results are illustrated in **Table 3**, we have the following observations:

- Overall, from the table, we can see that our proposed model consistently shows strong performance across all datasets, which ascertains our proposed method's effectiveness. Especially on the Amazon dataset, we note that existing baselines have already obtained high enough performance, while our approach still pushes that boundary forward.

---

[3]https://meta.wikimedia.org/wiki/Data_dumps.
[4]http://snap.stanford.edu/data/
[5]https://github.com/shawnwang-tech/GeniePath-pytorch

**Table 3: Comparison of different anomaly detectors in four real-world datasets and all the results are in percentage (%).**

| Methods | | Reddit | Alpha | Wikipedia | Amazon |
|---|---|---|---|---|---|
| | GCN | 72.2±1.4 | 84.8±0.9 | 71.7±2.1 | 83.7±1.2 |
| | GIN | 70.8±2.2 | 81.2±1.6 | 70.7±0.9 | 79.2±1.1 |
| | GraphSAGE | 68.9±3.0 | 85.4±2.1 | 71.4±2.5 | 77.3±2.8 |
| GNNs | GAT | 71.5±1.3 | 82.5±1.8 | 71.3±2.2 | 82.5±2.5 |
| | GeniePath | 72.4±3.0 | 87.3±2.9 | 72.5±1.4 | 80.8±1.3 |
| | Semi-GNN | 72.6±2.2 | 85.4±1.4 | 72.6±1.8 | 76.6±0.8 |
| | CARE-GNN | 72.5±3.1 | 83.2±3.7 | 73.4±2.5 | 84.7±2.8 |
| | ADMoE | 73.2±1.8 | 85.3±1.2 | 72.9±1.7 | 83.9±1.3 |
| | GAE | 72.6±2.3 | 86.5±2.5 | 72.1±2.1 | 79.5±2.8 |
| | DGI | 73.6±1.3 | 88.3±1.6 | 73.6±1.1 | 84.2±1.7 |
| SSL | DCI | 73.8±2.2 | 88.6±2.0 | 75.7±2.7 | 81.6±2.1 |
| | GRADATE | 72.7±1.4 | 89.1±1.8 | 73.2±2.1 | 83.8±2.7 |
| | ACT | 73.7±2.1 | 87.1±2.3 | 74.2±3.0 | 84.8±1.5 |
| | **Ours** | **76.91±1.1** | **92.3±2.2** | **78.6±0.9** | **89.9±1.8** |

- Compared with GNN-based methods, our model has achieved remarkable performance. We make the conjecture that we disentangle the latent factors of each node embedding and use the disentangled node representation to represent node capsules, which can preserve more rich and essential information, leading to better performance.

- Considering that GAE, DGI, DCI, GRADATE, ACT and our method all have the SSL architecture, we think the improvements mainly derive from the different SSL schemes. Our CapsGI is to learn higher-level representation for neighbors, such that the constative learning can only be performed between each node and the whole graph representation, which improves contrastive learning efficiently. As for other SSL schemes, whose self-supervised signal may not be obvious and significant.

## 5.3 Ablation Study

This section conducts experiments on four datasets to investigate the contribution of each component in our model. Specially, we design three variant versions of CapsGI:

- **R-DNC**: We remove the disentangled node capsule module when conducting node representation.

- **DGI**: We remove graph capsule infomax and instead use DGI for SSL task.

- **R-SSL**: We remove self-supervised learning task.

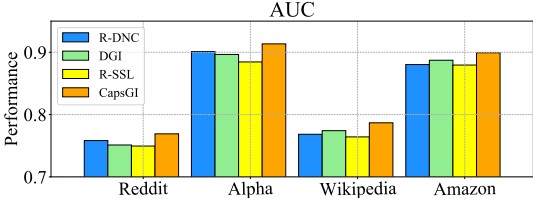

**Figure 4: Ablation study.**

To compare them under different conditions, we report their performance under the AUC metric. As shown in **Figure 4** , we can observe that each component consistently contributes.

- We disentangle the latent factors of each node embedding to preserve more rich and essential information. When removing this component, we can observe a lower performance drop on the AUC metric, demonstrating the effectiveness of disentangling concept. *(The essential components (**P1**) of our CapsGI.)*

- Besides, self-supervised learning improves the base model essentially, serving as the driving force of performance improvement. When removing the self-supervised task, we can observe a remarkable performance drop in the AUC metric, which indicates that SSL schemes can strengthen the connection among nodes. *(The essential components (**P2**) of our CapsGI.)*

- Moreover, we notice that our CapsGI can achieve better performance than DGI. The main reason is that our adaptive routing mechanism can select appropriate node capsules, so that favorable node capsules (e.g., normal users) can strengthen the connection against the fraudsters.

- Overall, CapsGI can outperform all of its variants, indicating the superiority of its design for anomaly detection.

## 5.4 Inconsistent Issue Experiments

Recall back **Figure 1**, an example of spam reviews detection on the graph, where the inconsistency between the behavior patterns and the label semantics is illustrated. Such inconstancy increases the difficulty of anomaly detection due to the wobbly users (e.g., wobbly users with inconsistency between labels and behaviors are hard to distinguish). In this section, we explore how the CapsGI training performs when the learning difficulty is varied (e.g., the number of wobbly users). To answer this question, we design the experimental protocol and present the observed results.

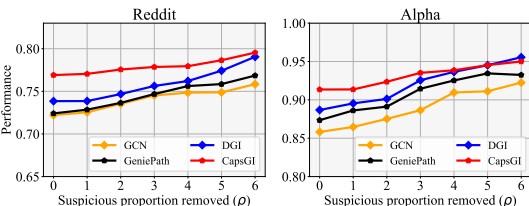

**Figure 5: The impact of various learning difficulties.**

**Experimental protocol.** We first train the GNN encoder and classifier and obtain the predicted suspicious score by using all the labeled users. Specifically, we conduct the training for 100 epochs and use each user's average predicted suspicious score. Then we sort the suspicious score that quantifies how anomalous each user is in descending order:

$$Score = \frac{\sum_{i=1}^{N}(Predict \rightarrow fraudsters)}{N}, N = Epoch \quad (16)$$

The top $\rho(\%)$ of suspicious scores in normal users and the bottom $\rho(\%)$ of suspicious scores in fraudsters are viewed as wobbly users with the inconsistency issue. Adjusting the value of $\rho$ and removing the corresponding wobbly users during classification will generate various learning difficulties levels. We set different learning difficulties via controlling the value $\rho$ in {0, 1, 2, 3, 4, 5, 6}, and choose two datasets as examples (e.g., Reddit, and Alpha).

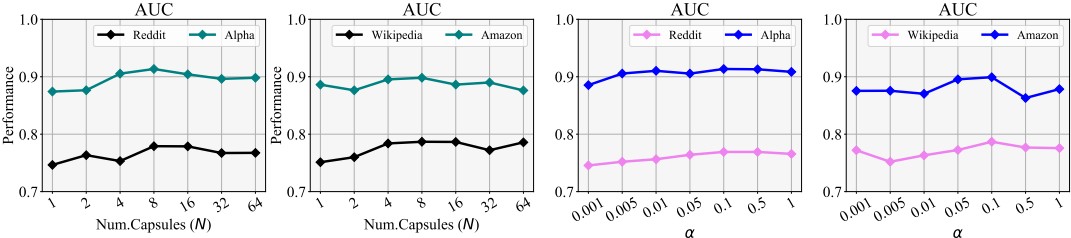

Figure 6: The impact of two hyperparameters.

**Observation.** As exhibited in **Figure 5**, when datasets have more wobbly instances (a smaller $\rho$), our proposed CapsGI brings more performance gains over the other baselines. While the dataset becomes easier to learn (a larger $\rho$), CapsGI only obtains comparable performance with the baselines. That is to say, besides the inconsistency, the effectiveness of anomaly detection could be impacted by other factors, such as scarce labels, which deserve further study.

## 5.5 Scalability Analysis.

Table 4: Scalability Analysis of disentangling ideas (%).

| Methods | Disentangling | Reddit | Alpha | Wikipedia | Amazon |
|---|---|---|---|---|---|
| GCN | No | 72.21 | 84.82 | 71.71 | 83.75 |
| | Yes | 73.42 | 84.98 | 73.28 | 84.52 |
| GeniePath | No | 72.41 | 87.35 | 72.69 | 80.88 |
| | Yes | 73.18 | 87.98 | 73.56 | 82.51 |
| DGI | No | 73.68 | 88.31 | 73.67 | 84.23 |
| | Yes | 74.18 | 89.15 | 75.11 | 85.02 |
| Ours | No | 75.83 | 90.12 | 77.84 | 88.36 |
| | Yes | 76.91 | 91.35 | 78.68 | 89.92 |

*Can disentangling ideas bring gains to other baselines?* To answer this question, we add the disentanglement block to three other GCN methods, i.e., GCN, GeniePath, and DGI, as examples. **Table 4** shows the performance for disentangling ideas employed in other baselines. In particular, We observe that adding the disentangle module brings different degrees of improvement to the model, demonstrating the disentangling module's effectiveness. Furthermore, it can be noticed that even if disentangling ideas are removed, our model still maintains remarkable performance. We can conclude that disentangling ideas is a node representation enhancement module with good scalability.

## 5.6 Model Parameter Analysis.

### 5.6.1 The impact of the number of higher-level graph capsules. We further explore the sensitivity of the number of higher-level graph capsules in four datasets. We summarize the results in **Figure 6** by ranging the number of capsules within $\{1, 2, 4, 8, 16, 32, 64\}$. We observe that the 8 capsules setting achieves the best performance for all datasets. Additionally, we notice that performance will not drop significantly when the number increases. Thus, we conclude that our approach is not very sensitive to this parameter.

### 5.6.2 The impact of hyperparameters $\alpha$. We have another hyperparameter to control the magnitude of the SSL tasks, i.e., $\alpha$. To

investigate the influence of $\alpha$, we report the performance with a set of representative $\alpha$ values in $\{0.001, 0.005, 0.01, 0.05, 0.1, 0.5, 1\}$ on four datasets. According to the results presented in **Figure 6**, the anomaly detection task achieves decent gains when jointly optimized with the self-supervised task. With the rise of $\alpha$, the performance increases first and then declines. We think it is due to the gradient conflicts between the two tasks. Besides, when $\alpha = 0.1$, we get the best performance.

## 5.7 Backbone Analysis.

Table 5: Backbone Analysis of our encoder (%).

| Backbone | Reddit | Alpha | Wikipedia | Amazon |
|---|---|---|---|---|
| GCN | 75.68 | 90.48 | 77.89 | 88.68 |
| GAT | 76.08 | **93.01** | 77.91 | 89.25 |
| GraphSAGE | 76.82 | 91.52 | 76.58 | 87.68 |
| GIN | **76.91** | 92.35 | **78.68** | **89.92** |

*Can different backbones of our graph encoder affect our performance?* To answer this question, we compare our model with four backbones, i.e., GCN, GAT, GraphSAGE, and GIN, on four datasets. **Table 5** demonstrates the performance for GapsGI employing four backbones. It can be noticed that all four backbones have achieved good results, especially the GIN backbone. And we can conclude that our CapsGI allows various choices of GNN architecture without any constraints. Additionally, we use GIN as the backbone, which can significantly boost the performance of CapsGI.

## 6 CONCLUSION

This paper proposes a new graph SSL scheme, Capsule Graph Infomax (called CapsGI), which inherits the strength of DGI for anomaly detection. To mine the intrinsic properties of each node, we disentangle the latent factors of each node embedding and use the disentangled node representations to represent node capsules. To strengthen the connection between normal nodes, the proposed CapsGI further characterizes the part-whole contrastive learning between lower-level capsules and higher-level capsules by explicitly considering the context graph relation. Extensive experiments show the superiority of the proposed model over the current advanced methods. Meanwhile, the research of capsule ideas for anomaly detection remains in its infancy, and their application in graphs has potential development, which is worthy of our further exploration.

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

## A  WHY CHOOSE CAPSULE REPRESENTATION?

We have the following three aspects. First, capsules use groups of neurons (named capsules) to represent features. Each capsule forms a local feature and represents one visible entity, such as a mouth or a nose. Therefore, capsules can retain more rich information. Second, by the dynamic routing algorithm, each lower-level capsule votes for only one higher-level capsule, such as the mouth that belongs to a face. Thus capsule representations make the network especially appealing in reasoning the part-whole hierarchy and robust to adversarial attacks. Such an advantage can be reminiscent of the contrast learning between the nodes (local) and the graphs (global) of CapsGI, which is very interesting. To the best of our knowledge, we are the first to combine capsules with SSL to conduct anomaly detection on graphs. Third, we proposed a novel dynamic routing mechanism to select appropriate node capsules adaptively. Such a mechanism is that only favorable node capsules (e.g., normal users) are gathered, and redundant node capsules (e.g., fraudsters) are restrained to better capture the whole graph structure.