# OpenReview forum: "Friend or Foe? Mining Suspicious Behavior via Graph Capsule Infomax Detector against Fraudsters"
_ACM.org/TheWebConf/2024/Conference — TheWebConf24 Oral_

### Official Review · Reviewer_fT88 · 2023-11-09

**Novelty:** 5
**Technical Quality:** 6

**Review:**

This paper presents a method to detect fraudulent users on the Internet. The main motivation of this paper is that fraudulent users often mimic ordinary users so that their features and interactions behave similarly, but their labels are different. On the other hand, ordinary users may behave very differently but have the same label. The authors propose a disentangled feature learning method to separate different factors in user behaviors, and also propose a capsule voting method to merge small capsules into larger ones. In addition, a contrastive learning method is proposed to mitigate the lack of labels. Extensive experiments show the effectiveness of the proposed CapsGI.

# Strong points.
1. I find the paper clearly organized and easy to follow.
2. The paper tackles an important real-world problem about the web. Fraudulent user detection is important for almost all online social networks.
3. I find the motivation of this paper to be clearly stated and also empirically verified. It is true that fraudulent users will connect to ordinary users to mimic their behaviors, which make the fraudulent detection problem difficult. The authors clearly state the motivation with figures, which facilitates understanding. More importantly, there is a dedicated section, Section 5.4, that is devoted to verify the motivation. This makes the paper more self-contained and enhances the motivation.
4. Experiments are done on four real-world datasets (which is extensive), and the improvements achieved are significant (over 3% improvement). Also, standard deviations are given.

# Weak points
1. From my perspective, what the authors do in Section 4.1 is not what people understand about 'disentangled' representation. In general, disentangled representation requires that each part of the features captures a unique aspect of the data, and that different parts of features  should be independent of each other. Furthermore, affecting one aspect of the data should not affect disentangled features for other aspects (Higgins et al. 2018). However, I fail to see any of these properties in Section 4.1. In fact, what the authors do resembles multi-head attention, in which different projection matrices are used, but no specific requirements of these projections are imposed. I feel that the disentanglement is somehow an overclaim here.
2. From my perspective, the authors fail to clearly show the different between the proposed capsule voting and existing graph pooling methods. The process of merging fine-grained node features to clusters and then to graph features resembles DiffPool (Ying et al. 2018) or StructPool (ICLR 2020). Maybe the authors would like to clarify the different between the proposed techniques and related pooling techniques.
3. This is a minor point, but in Section 5.5, "scalability" does not seem to be correctly used. Scalability commonly refers to how a method performs when the scale of the input data gets large (e.g. from 100 nodes to 1000, 10000, 100000, etc.)

(Higgins et al. 2018) Towards a Definition of Disentangled Representations. arXiv 1812.02230

(Ying et al. 2018) Hierarchical Graph Representation Learning with Differentiable Pooling. NeurIPS 2018.

(Yuan et al. 2020) Structpool: Structured graph pooling via conditional random fields, ICLR 2020.

**Questions:**

Q1, Q2 corresponds to Weaknesses 1 and 2.

Q3: I have a question regarding the design of the contrastive learning objective. In my opinion, simply shuffling rows of node features is not sufficient in this work. As this work specifically mentions "disentangled features", I would expect that some contrasts are done on the disentangled aspects. Any ideas about why this is missing?

**Ethics Review Description:**

Not needed.

**Reviewer Confidence:**

3: The reviewer is confident but not certain that the evaluation is correct

**Scope:**

4: The work is relevant to the Web and to the track, and is of broad interest to the community

---

### Official Review · Reviewer_k3B9 · 2023-11-23

**Novelty:** 6
**Technical Quality:** 6

**Review:**

This contribution proposes a novel graph anomaly detection method based on a graph neural network with capsule. The authors show that their method is more performant than previous method at detecting banned users in several social online datasets.

*Strengths:*
- New method using the recent idea of graph capsule.
- High performance on benchmark datasets.
- Detailed evaluation of the method, with the inconsistent issue experiments and ablation study.

*Weaknesses:*
- Some parts of the manuscript are not very clear. For example, it's not clear what the authors mean when they say that fraudsters can obtain information by GNNs as camouflage.
- The benchmark datasets are not sufficiently well described. It's not mentioned how many users are labeled as fraudsters.
- The related work section does not cover other successful approaches not based on GNN (such as [1] and [2])
-


[1] Gao, J.; Liang, F.; Fan, W.; Wang, C.; Sun, Y.; and Han, J. 2010. On community outliers and their efficient detection in information networks. KDD ’10, 813–822. ACM.
[2] Gutiérrez-Gómez, L., Bovet, A. and Delvenne, J.C., 2020, April. Multi-scale anomaly detection on attributed networks. In Proceedings of the AAAI conference on artificial intelligence (Vol. 34, No. 01, pp. 678-685).

EDIT After authors' answers: I am satisfied with their answers.

**Questions:**

see above

**Ethics Review Description:**

No issues

**Reviewer Confidence:**

2: The reviewer is willing to defend the evaluation, but it is likely that the reviewer did not understand parts of the paper

**Scope:**

3: The work is somewhat relevant to the Web and to the track, and is of narrow interest to a sub-community

---

### Official Review · Reviewer_RBsJ · 2023-11-24

**Novelty:** 5
**Technical Quality:** 6

**Review:**

The authors propose a novel usage of capsule networks for anomaly detection in graph neural networks. First the authors disentangle node features into node capsules. Then dynamic routing is used essentially as a clustering algorithm. Finally, the authors use contrastive learning as a self-supervised learning method to identify the anomalous nodes in each cluster.

The paper has state of the art performance on several real-world datasets. It also proves that capsule networks can work well in GNNs as well as the image domain. The greatest contribution is using node capsules for clustering and contrastive learning instead of traditional GNN clustering/SSL methods.

**Questions:**

How long is the training time for this network? Does the dynamic routing method take significantly longer than other clustering methods? Have you tried comparing the clusters made from dynamic routing to clusters from other methods?

**Reviewer Confidence:**

3: The reviewer is confident but not certain that the evaluation is correct

**Scope:**

4: The work is relevant to the Web and to the track, and is of broad interest to the community

---

### Official Review · Reviewer_72R9 · 2023-11-24

**Novelty:** 7
**Technical Quality:** 7

**Review:**

(this review was subreviewed)

Authors explain how existing approaches for anomaly detection on graphs suffer from the inconsistency existing between users' behaviours and labels. To address this challenge, they introduce a novel learning framework (CapsGI) relying on capsule networks and mutual information maximization, and which goal is to compute representations of graph elements from both local and global perspectives. To evaluate the performance of their approach, authors conduct experiments for an anomaly detection task on 4 real-world networks and compare results against Graph neural network-based and self-supervised learning-based methods. They show how their approach achieves higher performance on all evaluated datasets and provide conjectures on the causes for these results. They further investigate these causes through an ablation study. Moreover, authors design a specific study to evaluate model's performance under several difficulty levels considering inconsistency. Finally, they measure the scalability of their disentangling block and explore different designs for model parameters.

## Strengths

S1: The work is of high quality. The overall challenge, the context in which it occurs, the research questions and the methodology to address them are clearly explained. Authors' claims are well supported by the evidence given through the paper. Overall, the analysis acknowledge the limits of the most recent deep-learning based approches and present new interesting perspectives to overcome them. In summary, the approach is sound and the methodology is robust.

S2: The experiments in the paper are extensive and convincing. The experimental results justify the effectiveness of the proposed method.

## Weaknesses

W1: Some formulations are difficult to understand and obscure the underlying message. For instance, in the P1 paragraph (introduction), with the sentence "In order to mine [...]". Or in the Figure 2, with the curved arrows subgraph above the "route (clustering)" block being difficult to understand. Just before subsection 5.7, authors mention "gradient conflicts between the two tasks" , which is not clear.

W2: No information is given about computation time required for the proposed approach. However, a serious increase in computation time could mitigate the performance gains displayed in Table 3.

W3: Source code for experiments and model development is not publicly released.

Minor remarks:
- In the P2 paragraph (introduction), SSL term is used but not defined. It is later used in the contribution highlight, with corresponding definition (Self Supervised Learning)
- In 4.1, just before equation (6), [... the presence of a node via using ...]
- Just before equation (8), [... hat receives]
- Figure 3 is really small and difficult to read

**Questions:**

See weaknesses above

**Ethics Review Description:**

--

**Reviewer Confidence:**

3: The reviewer is confident but not certain that the evaluation is correct

**Scope:**

4: The work is relevant to the Web and to the track, and is of broad interest to the community

---

### Decision · Program_Chairs · 2024-01-22

**Decision:**

Accept (Oral)

**Comment:**

The paper describes a novel graph anomaly detection technique based on GNN's with capsule.
 The writing was appreciated, and the experiments are convincing. The authors did a good job in answering to the reviewers' comments.
 I suggest that they incorporate in the final version the main points raised during the discussion.